:ᐧ᷍ PLOS | ONE

# Increased cell size, structural complexity and migration of cancer cells acquiring fibroblast organelles by cell-projection pumping

Hans Zoellner[1,2]*, Belal Chami[1], Elizabeth Kelly[1], Malcolm A. S. Moore[2]

**1** The Cellular and Molecular Pathology Research Unit, Oral Pathology and Oral Medicine, School of Dentistry, Faculty of Medicine and Health, The University of Sydney, Westmead Hospital, Westmead, NSW, Australia, **2** Cell Biology, The Memorial Sloan Kettering Cancer Center, New York, NY, United States of America

* hans.zoellner@sydney.edu.au

**Data Availability Statement:** Data has been made available at the University of Sydney Library repository and can be accessed via the following URL: https://protect-au.mimecast.com/s/

## Abstract

We recently described a hydrodynamic mechanism for cytoplasmic transfer between cells, termed cell-projection pumping (CPP). Earlier image analysis related altered SAOS-2 osteosarcoma cell morphology, to what we now recognize as CPP uptake of fibroblast cytoplasm. We here examine SAOS-2 phenotype following co-culture with human dermal fibroblasts (HDF) in which organelles were pre-labelled with a fluorescent lipophilic marker. Fluorescence activated cell sorting (FACS) analysis was performed of HDF and SAOS-2, cultured either alone or together. FACS forward scatter is proportionate to cell size, and increased for SAOS-2 with high levels of HDF fluorescence uptake (p < 0.004). FACS side scatter is proportionate to internal cell complexity, and increased in SAOS-2 with increasing uptake of HDF fluorescence (p < 0.004), consistent with uptake of HDF organelles. Scratch migration assays revealed that HDF migrated more quickly than SAOS-2 in both isolated cell culture, and following co-culture (p < 0.004). Notably, SAOS-2 with high levels of HDF labelling migrated faster compared with SAOS-2 with low HDF labelling (p < 0.008). A slight and unconvincing reduction in SAOS-2 proliferation was seen (p < 0.02). Similar results were obtained in single additional experiments with A673 and H312 cancer cells. Forward and side scatter results suggest organellar transfer by CPP increases cancer cell morphological diversity. This may contribute to histological pleomorphism relevant to cancer diagnosis and prognosis. Also, increased migration of sub-populations of cancer cells with high CPP organellar uptake, may contribute to invasion and metastasis in-vivo. We thus suggest relevance of CPP to cancer diagnosis and progression.

## Introduction

We earlier described the exchange of cytoplasmic protein and organellar membrane between cultured human fibroblasts and cancer cells (CC) [1]. Others have made similar observations, and describe this as via either tunneling nanotubes (TNT) or exosomes and other shed membrane vesicles, and this is often associated with changes in cell phenotype [2–22]. At the time

ZudxC81Zj6tRjwPPCnRFAX?domain=ses.library.
usyd.edu.au.

**Funding:** This work was supported by the
Memorial Sloan Kettering Cancer Center, including
via MSKCC P30 CA008748 Cancer Center Support
Grant; (https://www.mskcc.org) to MASM and
supported by the Australian Dental Research Fund
and Australian Dental Industry Association (https://
www.ada.org.au/AusDentResearchFoundation/
ADRF-Home) to HZ. An anonymous donor
supported the laboratory to HZ. The funders had no
role in study design, data collection and analysis,
decision to publish, or preparation of the
manuscript.

**Competing interests:** The authors have declared
that no competing interests exist.

of our earliest report, and in absence of time-lapse recordings, we assumed TNT likely responsible, and used the term 'cellular sipping' to convey our sense of cells sipping cytoplasm from one another [1]. However, our recent time-lapse recordings showed transfer in our co-cultures was not via either TNT or shed vesicles. Instead, transfer was by a mechanism not seemingly previously reported and for which we have proposed a hydrodynamic mechanism, 'cell-projection pumping' (CPP) [23].

Details of CPP are available elsewhere [23], but in brief, CPP as observed by time-lapse fluorescence microscopy was mediated by highly mobile and often branching cell-projections in the size range of filopodia, that writhed adherent to the culture surface and alternately probed and retracted from neighboring cells [23]. Although the rapid movement and small size of these cell-projections obscured precise visualization, they were clearly different to TNT, which have a straight morphology, change little over prolonged periods of time, and are suspended above the culture surface as taught wire-like connections [2–10, 12, 13, 24, 25]. Increased hydrodynamic pressure in retracting cell-projections, normally returns cytoplasm to the cell body. We suggest, however, that in CPP, cytoplasm in retracting cell-projections equilibrates partially into adjacent recipient cells via temporary inter-cellular cytoplasmic continuities. Although the precise mechanism for formation of these intercellular continuities is uncertain, precedent for such structures is established by the formation of TNT [6–8, 12, 13, 24–26]. Because pressure equilibrates preferentially towards least resistance, CPP transfer is affected by cell stiffness. We did observe some TNT in our time-lapse recordings, but transfer by CPP appeared quantitatively more significant, and this was supported by mathematical modelling and computer simulations [23]. The current study was to determine if cytoplasm uptake by CC in a culture system known to have predominant CPP, affects CC phenotype.

With regard to the method used to observe CPP, it is important to appreciate necessity to use permanent labels, such as the fluorescent lipophilic markers 1,1'-dioctadecyl-3,3,3',3'-tetramethylindodicarbocyanine perchlorate (DiD) and 3,3'-dioctadecyloxacarbocyanine perchlorate (DiO), to demonstrate total cytoplasmic transfer, because such labels persist long after degradation of the originally labelled structures. By contrast, cell turn-over renders highly specific organellar or protein labels unreliable for detecting cumulative cytoplasmic transfer between cells [1]. Both DiD and DiO mark organelles strongly, with negligible labelling of plasma membrane [1, 23]. In our earlier report, we used DiD and DiO to observe transfer of membrane structures, being primarily organelles, as well as the separate fluorescent markers CFSE and DDAOSE that label cytoplasmic proteins [1]. DiD and DiO were again used in more recent time-lapse microscopy, because transfer of punctate organellar labelling is more readily observed than diffuse cytoplasmic protein label in delicate cell-projections [23]. CPP transfer of lipophilic organellar and cytoplasmic protein fluorescent labels are comparable [1]. Nonetheless, preliminary experiments for the current work, showed more ready separation of sub-populations of cells when cytoplasmic transfer was determined using DiD and DiO organellar labels, as opposed to CFSE and DDAO-SE cytoplasmic protein fluorescence.

Assessment of CPP on CC morphology was on basis of the established relationship between forward and side scatter in Fluorescence activated cell sorting (FACS) analysis, and cell size and internal structural complexity respectively [27]. Further phenotypic characterization was based on the capacity of FACS to separate sub-populations of co-cultured cells [27].

## Materials and methods

### Materials

All culture media including M199, α-MEM, M5, Trypsin (0.25%)-EDTA (0.02%) and PBS, as well as Penicillin (10,000 U/ml)-Streptomycin (10,000 μg/ml) concentrate solution were

prepared and supplied by the Memorial Sloan-Kettering Cancer Centre Culture Media Core Facility (New York, NY). Amphoteracin B was purchased from Life Technologies (Grand Island, NY). Gelatin was from TJ Baker Inc (Philipsburgh, NJ). Bovine serum albumin was from Gemini Bioproducts (West Sacramento, CA). Falcon tissue culture flasks and centrifuge tubes were purchased from BDBiosciences (Two Oak Park, Bedford, MA). HDF were from The Coriell Institute (Camden, NJ). SAOS-2 osteosarcoma cells were from the American Type Culture Collection (VA, USA). A673 osteosarcoma and H3122 lung carcinoma cells were from the collection at the Memorial Sloan Kettering Cancer Center. The lipophilic fluorescent probes DiD (excitation 644nm, emission 665nm) and DiO (excitation 484nm, emission 501nm) Vybrant cell labelling solutions were from Molecular Probes, Life Technologies (Grand Island, NY). DAPI was provided by the FACS core facility at Memorial Sloan Kettering Cancer Center.

## Cell culture

The antibiotics penicillin (100 U/ml), streptomycin (100 μg/ml) and amphotericin B (2.5μg/ml) were used throughout all cell culture. Culture conditions differed according to cell type, such that: HDF were always cultured on gelatin coated surfaces (0.1% in PBS) in alpha-MEM (15% FCS); A673, were cultured in DMEM (10% FCS); the osteosarcoma cells SAOS-2 and U2OS were in M199 (10% FCS); and the lung adenocarcinoma line H3122 were in M5 (10% FCS) [1, 28]. Cells were harvested using trypsin-EDTA, into FCS to neutralize trypsin, and pelleted by centrifugation before passage at a ratio of 1 to 3. All cell culture was performed at 37˚C under $CO_2$ (5%) and at 100% humidity.

## Labelling of cells with lipophilic fluorescent membrane markers

Labelling solutions of DiD (1mM) and DiO (2mM) were prepared in alpha-MEM with 10% FCS, and applied to cells for 30 minutes in the case of DiD, while DiO was applied for 1 Hr. Cells were then washed twice with PBS before overnight culture with alpha-MEM with BSA (4%) followed by two further washes with PBS in order to ensure removal of any unbound label [1, 23, 28].

## Co-culture conditions

All experiments were performed with cells cultured on gelatin (0.1% in PBS) coated surfaces. Fibroblasts were seeded at from 1 to 2 x $10^4$ cells per $cm^2$ and allowed to adhere overnight before labelling with DiD and further overnight culture in alpha-MEM with BSA (4%) as outlined above. MC were seeded prior to labeling at near confluence in culture media appropriate to the MC line, and allowed to adhere overnight before labeling with DiO and further overnight culture in alpha-MEM with BSA (4%) as outlined above. MC pre-labelled with DiO where then seeded in alpha-MEM with BSA (4%) at a culture density of 4 x $10^4$ cells per $cm^2$ for 24 Hr co-culture. Control cultures comprised fibroblasts and MC seeded in parallel with or without labeling. HDF (pre-labelled with DiD) and MC (pre-labelled with DiO), co-cultured in 3 separate 175 $cm^2$ flasks before pooling for FACS separation. Separate in 25 $cm^2$ flask cultures of HDF and MC, both with and without fluorescent labelling, were prepared to establish FACS gates.

## FACS cell sorting separation

Labelling of HDF with DiD and co-culture with MC labelled with DiO provided the greatest possible fluorescent separation of MC with low as opposed to high HDF label. SAOS-2 were

used in most experiments based on favorable FACS fluorescence patterns, as well as because most data in earlier studies was from experiments with SAOS-2 [1, 29, 30]. In addition, one experiment was performed with H3122, and a further single experiment was performed with A673. Depending on the availability of instruments, cells were sorted using either a FACSAria-I or FACSAria-III cell sorter (Becton Dickinson & Co, Frankin Lakes, NJ).

Cells for FACS separation were harvested using trypsin-EDTA and pelleted in the presence of serum to neutralize trypsin, before suspension in from 300 μl to 500 μl of a resuspension solution comprising 50% PBS, 47% DMEM-alpha and 3% FCS with DAPI (0.1 μg/ml). Voltages and gates for separation of cells were established using: unlabeled HDF, HDF labelled with DiD only, unlabeled MC, MC labelled with DiO only, and MC labelled with DAPI only. In early experiments, HDF and MC cultured alone were combined for separation and assay identical to co-cultured cells, and no clear difference between HDF cultured alone or together with MC, or between MC with low HDF label and MC cultured alone was found. To reduce the potential influence of contaminant HDF in MC separated from co-cultures, in some later experiments FACS separation was performed in two steps, firstly separating HDF from MC, and then separating MC into those with high or low HDF labelling, and no clear effect of this was found on results. Post-sort analysis was performed on HDF and MC populations isolated from co-cultures, and in some experiments, additional controls were of MC otherwise cultured alone, but deliberately contaminated (spiked) with HDF at levels determined by post-sort analysis to be present in co-cultured MC with high HDF label. Cells obtained as outlined above were used for analysis of forward and side scatter, migration, and proliferation.

## Analysis of forward and side scatter

Flowjo software (FlowJo LLC, Ashland, OR) was used to prepare histograms of forward and side scatter for HDF cultured alone, MC cultured alone, HDF co-cultured with MC, and MC with high or low HDF labelling. Superimposition of histograms permitted visual comparison of differences in forward and side scatter between populations within individual experiments. To quantitate these differences, the position of peaks for forward and side scatter were identified for MC with low HDF labelling, and the relative percentage of cells with scatter greater than these peaks determined in all cell populations studied. The Wilkoxon Signed Rank Test in Prism 6.0e software was used to evaluate the statistical significance of differences between populations across multiple experiments with SAOS-2.

## Migration assay

Migration was quantitated using the scratch migration assay modified from that described by others [31]. In brief, felt-tipped pens were used to mark the under-surfaces of 48 well tissue culture plates with single vertical lines bisecting wells. Wells were coated with gelatin (0.1% in PBS) before seeding quadruplicate wells with: HDF or HDF cultured with MC separated by FACS, each at $1.5 \times 10^4$ cells per well in alpha-MEM with FCS (10%); or alternatively MC, MC with low HDF label separated by FACS, or MC with high HDF label separated by FACS, each at $3 \times 10^4$ cells per well, using M199 with FCS (10%) in the case of SAOS-2, or M5 with FCS (10%) when H3122 were studied, and DMEM with FCS (10%) when A673 were used. Cells were allowed to attach overnight, before using 1000 μl 'blue pipette tips' to inflict single horizontal scratches that bisected wells perpendicular to the pen marks scored on well under-surfaces. Phase contrast images of scratches were collected for up to 7 days using a 4 x objective in a Nikon Eclipse Ti inverted phase contrast microscope with NIS Elements F3.0 software (Nikon Corporation, Tokyo, Japan). It was possible to reliably photograph the same scratch sites throughout entire observation periods, using the pen marks perpendicular to the scratches

for localization. The full width of scratches was visible in photomicrographs, so that the advancing front of migrating cells approaching each other was readily identified in sequential images over time. Adobe Photoshop CS6 software was used to prepare black and white images in which the scratch surface areas bounded by the opposing advancing fronts of cells, were marked white against a black background comprising surfaces covered by cells. Image J software was then used to calculate scratch surface areas, which were expressed in terms of $\mu m^2$ as determined from photomicrographs of a haemocytometer. Subtraction of scratch surface areas from later time points from initial scratch surface areas at 0 Hr, permitted calculation of the average distance migrated by cells at each site and time point monitored. Statistical evaluation of multiple experiments with SAOS-2 was by the Wilkoxon Signed Rank Test in Prism 6.0e software.

### Proliferation assay

Proliferation of cells was assessed by monitoring gentian violet staining over time, using a method modified from that described by others [32, 33]. In brief, 96 well plates were coated with gelatin (0.1% in PBS) before seeding quadruplicate wells with: HDF or HDF cultured with MC separated by FACS, each at $0.5 \times 10^4$ cells per well in alpha-MEM with FCS (10%); or alternatively MC, MC with low HDF label separated by FACS, or MC with high HDF label separated by FACS, each at $1 \times 10^4$ cells per well, using M199 with FCS (10%) in the case of SAOS-2, or M5 with FCS (10%) when H3122 were studied, and DMEM with FCS (10%) when A673 were used. 96 well plates were seeded in such a way that there was one plate containing all cell populations tested for each time point studied. Cells were allowed to adhere overnight before fixation of what was designated the zero hr time point by first discarding medium and applying paraformaldehyde (4% in PBS) for 30 min, washing five times under tap water and air drying. Once plates representing all time points had been fixed, washed and air dried as described above for the zero hr time point, wells with cells and additional blank wells, were stained with 100 μl volumes of Gram gentian violet solution for 1 Hr. Plates were then washed five times under running tap water before again air drying. Acid alcohol (50% ethanol, 50% 0.1M HCl) was used to solubilize stain before measurement of optical density at 450 nm using a Synergy H1 Hybrid Reader and Gen5 2.00 Software from BioTek (Winooski, VT), before exporting results to Excel. The mean optical density was determined for blank wells at each time point, and subtracted from individual optical density readings for wells containing cells, before calculation of mean optical density for each quadruplicate. Changes in mean optical density of cultures over time, were expressed as percentages relative to a value of 100 assigned to the mean optical density of the zero hr time point, and this was assumed to represent proportional change in cell number [32, 33] in Prism 6.0e software was used to evaluate statistical significance across multiple experiments with SAOS-2.

### Ethical considerations

Experiments were with commercially sourced HDF and widely propagated CC lines, so no direct ethical concerns arise.

## Results

### Forward and side scatter of cc increased on uptake of fibroblast fluorescent label

Organellar transfer during co-culture generated two readily separable populations of CC, being those with high as opposed to low levels of HDF labelling (Fig 1A). Forward scatterplots confirmed that control HDF cultured alone had greater forward scatter compared with SAOS-

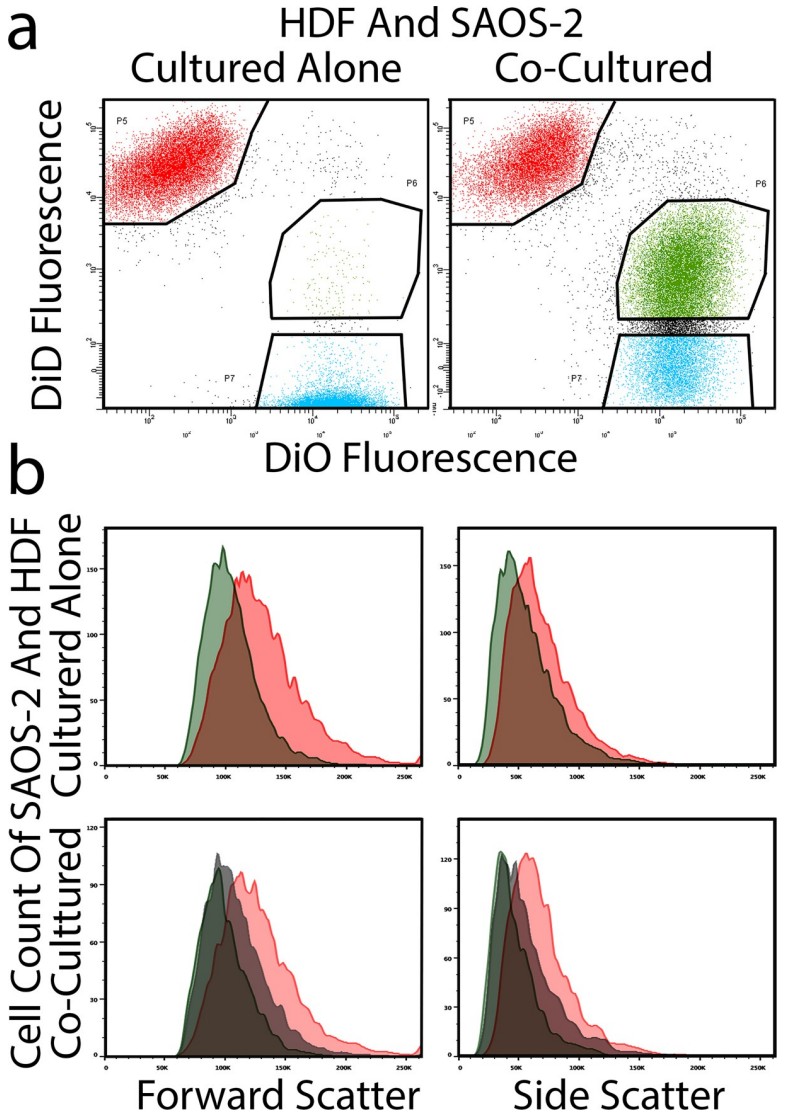

**Fig 1. Cell size and internal complexity analyzed by FACS.** FACS plots are shown of HDF and SAOS-2 cultured alone or together (a), as are histograms of forward and side scatter of defined populations (b). **(a)** Good FACS separation of HDF (red) from SAOS-2 (blue) was seen when cells were cultured alone. Co-culture resulted in appreciable transfer of HDF fluorescence to SAOS-2. Gates are indicated for definition and sorting co-cultured cells into HDF co-cultured with SAOS-2, SAOS-2 co-cultured with HDF and with high HDF label (green), or SAOS-2 co-cultured with HDF and with low HDF label (blue). **(b)** Representative histograms are shown for forward and side scatter of HDF and SAOS-2 cultured alone (HDF, red; SAOS-2, green), or after 24 h co-culture (HDF co-cultured with SAOS-2, red; SAOS-2 co-cultured with HDF and with low HDF label, green; and SAOS-2 co-cultured with HDF and with high HDF label, grey). HDF had appreciably right-shifted forward and side scatter compared with SAOS-2, both when cultured alone and in co-culture. SAOS-2 co-cultured with HDF and with high HDF label, had right-shifted forward and side-scatter, compared with fellow SAOS-2 with low levels of HDF labelling.

2 in similar isolated culture. This relationship was maintained following co-culture (Fig 1B). Notably, SAOS-2 with high levels of HDF labelling had greater forward scatter compared with those with low levels of HDF labelling (Fig 1B). Similar results were obtained for FACS side scatter (Fig 1B). These observations were reproduced across nine separate experiments co-culturing HDF with SAOS-2 (Fig 2; $p < 0.008$, Wilkoxon Signed Rank Test), while similar changes were seen in one experiment each with H3122 and A673 cell lines (Fig 2).

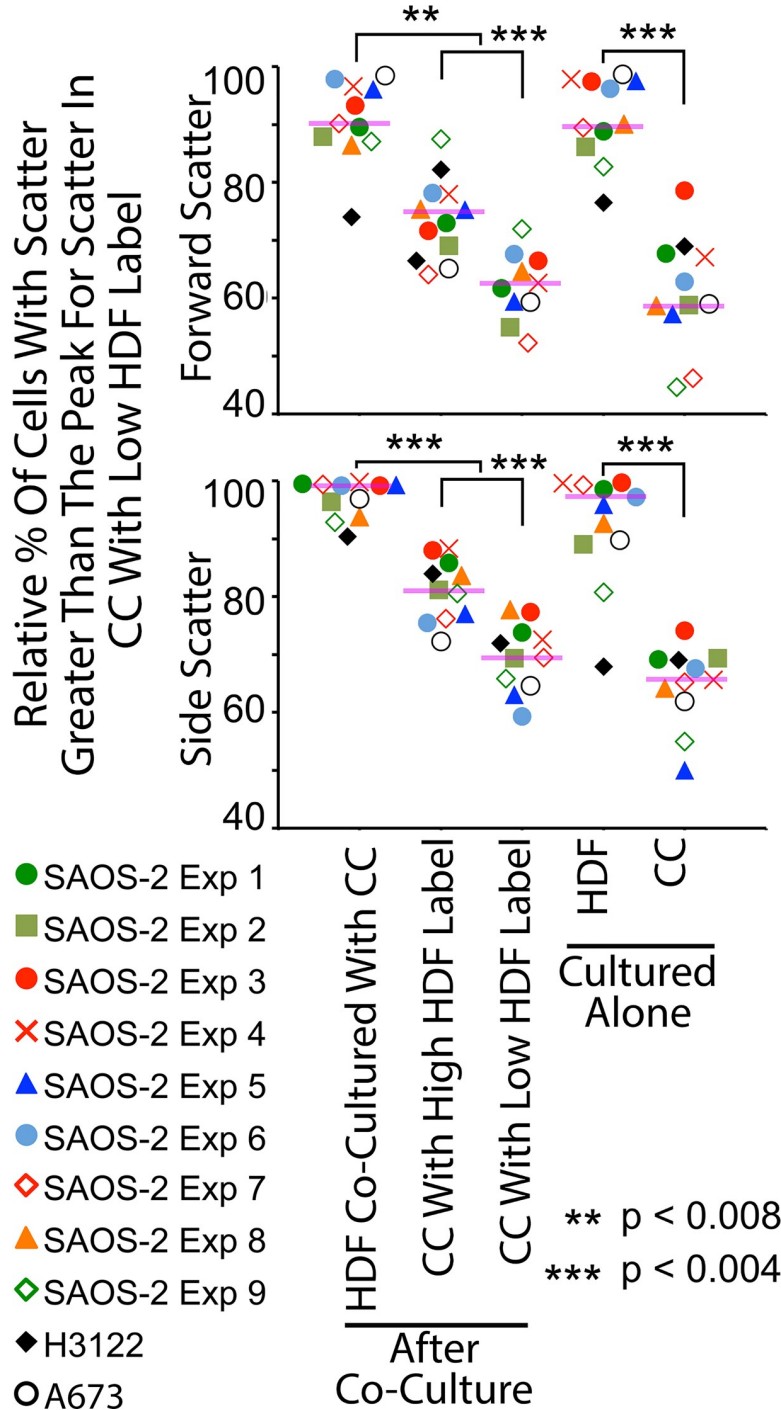

**Fig 2. Scattergrams of FACS forward and side-scatter across multiple experiments.** Results shown are of 9 experiments with SAOS-2 (colored markers), and one experiment each with A673 (black open circles) and H3122 (black diamonds). Purple horizontal lines indicate median values for SAOS-2, and statistical significance for SAOS-2 is as shown (Wilcoxon Signed Rank Test, **p < 0.008, ***p <0.004).

## Uptake of fibroblast organelles by CPP increased cc migration but had little effect on CC division

FACS separation of the cells within the gates indicated in Fig 1A, permitted study of the effect of fibroblast organelles transferred to CC on cell migration. Fig 3A shows that control HDF migrated appreciably faster compared with control SAOS-2. The comparatively fast migration of HDF was also seen when cells were isolated by FACS separation from co-cultures (Fig 3B). Notably, however, co-cultured SAOS-2 with high levels of HDF labelling migrated faster compared with SAOS-2 with low levels of HDF labelling. Also, deliberate contamination of otherwise pure SAOS-2 cultures with HDF at levels revealed by post-sort analysis to contaminate FACS separated SAOS-2 from co-cultures, had no significant effect on migration (Fig 3A). This was seen in nine separate experiments with SAOS-2 (p < 0.008, Wilkoxon Signed Rank Test), and in one further experiment with A673 cells (Fig 4).

We also examined the proliferation of these cell populations (Fig 5). As expected, SAOS-2 proliferated at a greater rate compared with HDF (p < 0.02, Wilkoxon Signed Rank Test). A statistically significant but otherwise unconvincing reduction in median cell number was seen in SAOS-2 with high as opposed to low levels of HDF labelling (Fig 5, p < 0.02, Wilkoxon Signed Rank Test).

## Discussion

Given suspension of organelles in cytoplasm, as well as our earlier observations of both membrane organellar and cytoplasmic protein transfer by CPP [1, 23], it is reasonable to consider DiD organellar transfer in the current study as correlated with total cytoplasm transfer, and as a measure of CPP. Although the precise identity of organelles transferred is not established, earlier microscopy showed transfer of organelles ranging from those at the limit of visual resolution, through to much larger organelles with the size of mitochondria [1, 23], and we have separately observed transfer of labelled mitochondria via CPP.

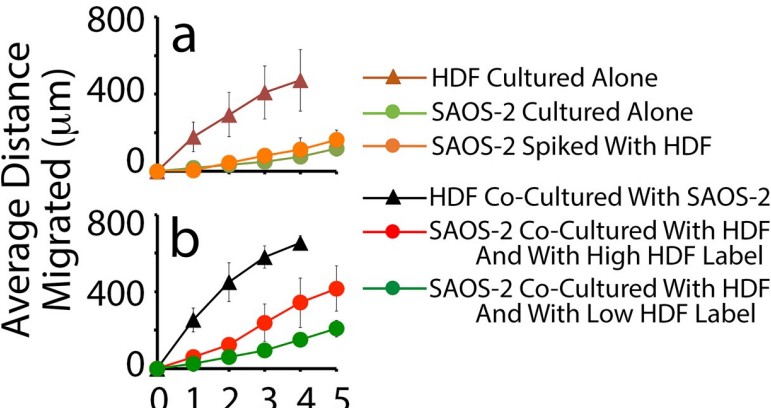

**Fig 3. Time course of migration in scratch assays.** Cells examined were: HDF and SAOS-2 cultured alone; SAOS-2 spiked with HDF at levels known to contaminate SAOS-2 co-cultured with HDF and isolated by FACS; HDF co-cultured with SAOS-2; and SAOS-2 co-cultured with HDF and with either low or high levels of HDF labelling (mean of 4 experiments). (**a**) Control HDF cultured alone migrated appreciably faster compared with control SAOS-2 cultured alone. SAOS-2 isolated from co-cultures with HDF by FACS, had between 1% and 2% contamination with HDF. Spiking SAOS-2 cultures with HDF at these levels had no detectable effect on SAOS-2 migration. (**b**) HDF isolated by FACS following 24 h co-culture, also migrated at a greater rate compared with SAOS-2 with which they had been cultured. Notably, SAOS-2 with high levels of HDF labelling migrated at an appreciably greater rate compared with SAOS-2 with low levels of HDF labelling.

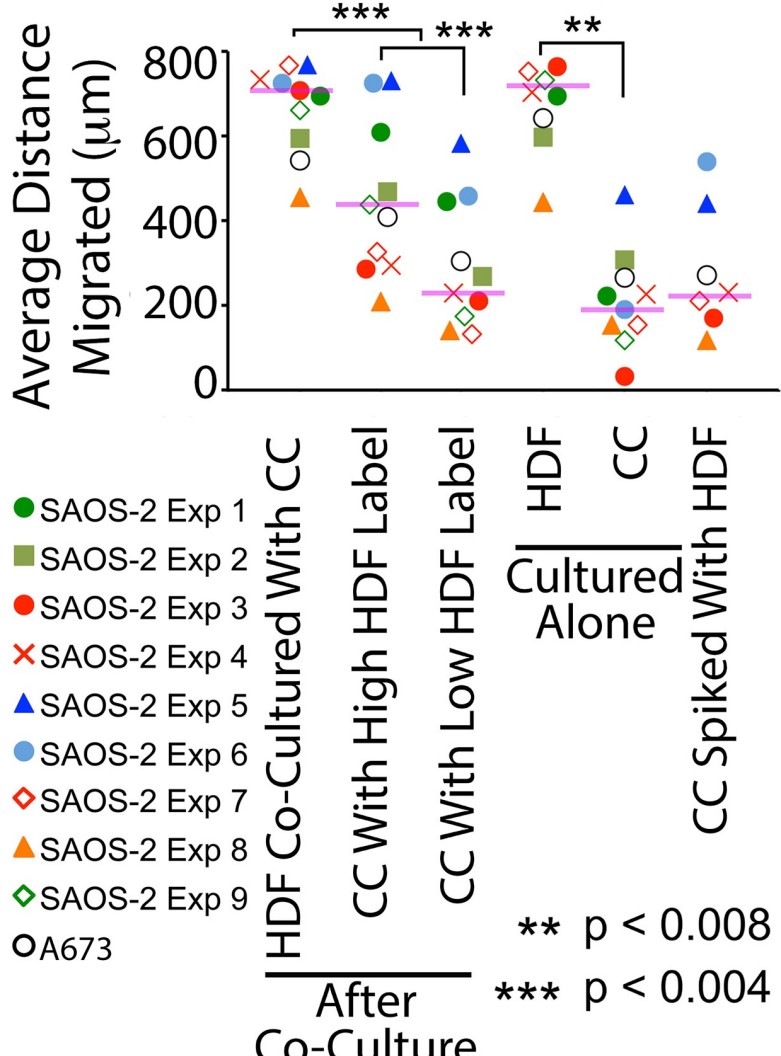

**Fig 4. Scattergram of scratch migration assay results across multiple experiments.** Results shown are of 9 experiments with SAOS-2 (colored markers), and one experiment each with A673 (black open circles). The same SAOS-2 experimental codes are used in this scattergram as in Figs 2 and 5, to permit direct comparison. The differences in migration seen in the time-courses illustrated in Fig 3, were reproducible across 9 experiments with SAOS-2 (colored markers), and in one experiment with A673 (black open circles). Purple horizontal lines indicate median values for SAOS-2, and statistical significance for SAOS-2 is as shown (Wilkoxon Signed Rank Test, **p < 0.008, ***p <0.004).

While CPP remains distinct from cytoplasmic transfer via TNT [23], the presence of inter-cellular cytoplasmic continuities in both suggest these two forms of intercellular transfer are more similar to each other, than either are to exosome mediated transfer. Importantly, exosomes do not contribute significantly to cytoplasmic fluorescence transfer in SAOS-2 –HDF co-cultures, as evidenced by: absence of fluorescence transfer by conditioned media; highly localized fluorescence transfer to individual cells; and time-lapse microscopy showing discrete transfer events and transwell membrane experiments [1, 23, 30].

FACS forward scatter is proportionate to cell size [27], so that increased SAOS-2 forward scatter in the current study, demonstrates a role for CPP in increasing CC size. Similarly, FACS side-scatter correlates with internal structural complexity of cells [27], and this is

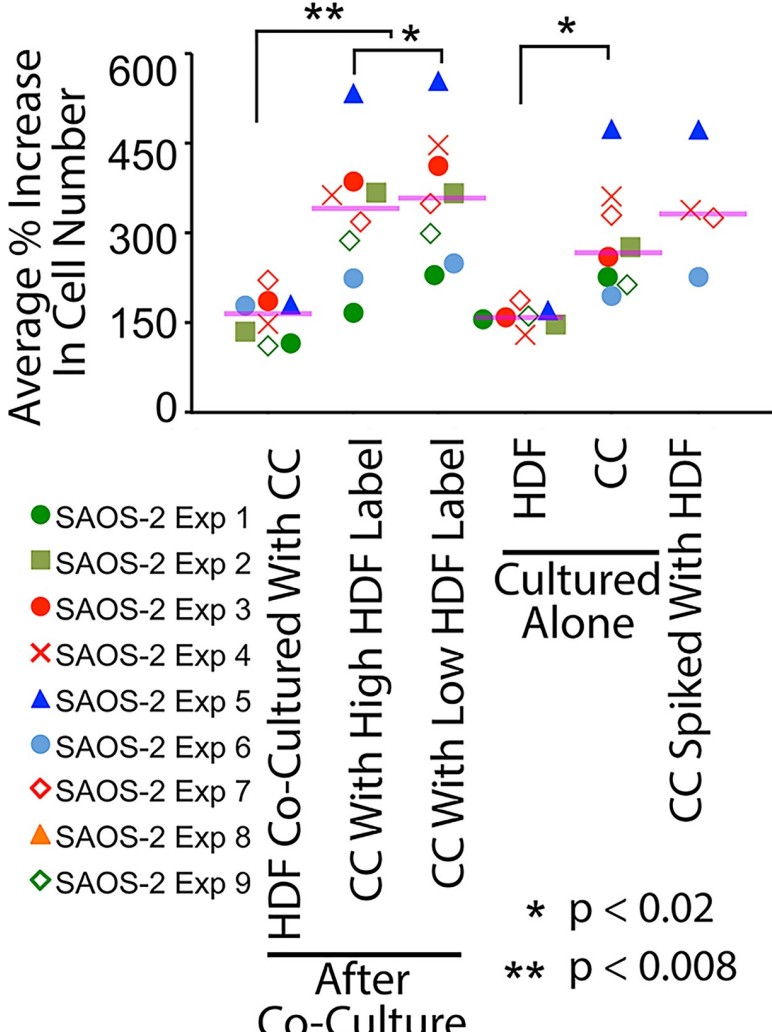

**Fig 5. Scattergram of proliferation assay results across multiple experiments.** Results shown are of 9 experiments with SAOS-2. The same SAOS-2 experimental codes are used in this scattergram as in Figs 2 and 4, to permit direct comparison. Purple horizontal lines indicate median values. SAOS-2 were more proliferative compared with HDF, while a slight but statistically significant difference in proliferation assay results was seen between co-cultured SAOS-2 with high as opposed to low HDF labelling. Spiking SAOS-2 with HDF at levels found by post-sort analysis to contaminate co-cultured SAOS-2 separated by FACS, did appear to affect the proliferation assay. However, there was an insufficient number of replicates for reliable statistical evaluation of this result. The small magnitude of the difference seen in proliferation between co-cultured SAOS-2 with high compared with low HDF label uptake, coupled with possible confounding of results by halving of acquired fluorescence during cell division, undermine confidence in the possible modest effect seen. For these reasons, we do not report a clear effect of organellar transfer of HDF organelles to SAOS-2 on SAOS-2 proliferation. Statistical significance is as shown (Wilkoxon Signed Rank Test, **p < 0.02, ***p < 0.008).

consistent with the appreciable transfer of HDF organelles expected from separate time-lapse recordings CPP [23]. Because CPP generates sub-populations of CC with altered morphology, we suggest that the increased morphological diversity seen in the current data may represent a role for CPP in generation of CC morphological diversity, and hence in histopathological pleomorphism relevant to cancer diagnosis and prognosis [34]. This is consistent with our earlier report of altered SAOS-2 cell profile area and circularity following CPP [1].

It is noteworthy that although CPP may be more akin to intercellular exchange via TNT than via exosomes, that we are unable to find any reports of altered cell morphology following

uptake of material via CPP, while there are several separate reports of altered cell morphology following exosome uptake [14–16]. Similarly, while it has been long known that exosomes from a variety of sources can increase migration of several cell types [17–21], there is less evidence for a similar effect for cellular contents transferred via TNT. Our observation of increased SAOS-2 migration following fibroblast cytoplasmic CPP, is consistent with a recent report of increased cancer cell migration after uptake of mitochondria via TNT from macrophages [22]. We cannot, however, attribute increased migration in our data to mitochondria alone, because CPP transfers bulk cytoplasm with an admixture of organelles and cytoplasmic proteins [1, 23]. Seemingly different effects of cytoplasmic transfer by CPP and exchange via TNT, underscore the distinction between the two processes.

Although we did observe a statistically significant effect of organellar transfer on SAOS-2 proliferation, we were unconvinced by the slight apparent change in proliferation. This is because cell division halves cell fluorescence, 'down-shifting' rapidly dividing cells in FACS plots. By contrast, we are more confident interpreting increased migration after CPP, because there was a large increase as opposed to small decrease, in the phenotypic characteristic measured.

It is noteworthy that scratch migration assays were over many days, and that increased CC migration persisted despite ongoing cell division. We speculate that this persistence reflects epigenetic effects of CPP, but confirmation or otherwise of this possibility demands further experimentation. It seems probable that cells other than fibroblasts would have similar exchange with CC, and we have observed this in preliminary experiments with smooth muscle cells. The molecular basis for CPP requires separate investigation, but based on the current data, inhibition of exchange could reduce invasion and metastasis, offering new therapeutic targets for cancer treatment.

## Acknowledgments

We thank the Memorial Sloan Kettering Cancer Center, including via MSKCC P30 CA008748 Cancer Center Support Grant, as well as the Australian Dental Research Fund and Australian Dental Industry Association for their support of this work. We also thank an anonymous donor for their generous contribution.

## Author Contributions

**Conceptualization:** Hans Zoellner.

**Data curation:** Hans Zoellner.

**Formal analysis:** Hans Zoellner.

**Funding acquisition:** Hans Zoellner, Malcolm A. S. Moore.

**Investigation:** Hans Zoellner, Elizabeth Kelly.

**Methodology:** Hans Zoellner.

**Project administration:** Hans Zoellner.

**Resources:** Hans Zoellner, Malcolm A. S. Moore.

**Writing – original draft:** Hans Zoellner.

**Writing – review & editing:** Hans Zoellner, Belal Chami, Elizabeth Kelly, Malcolm A. S. Moore.

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
