## [Decision Letter · Decision Letter 0]

23 Oct 2019

Increased cell size, structural complexity and migration of cancer cells acquiring fibroblast organelles by cell-projection pumping

PONE-D-19-21420

Dear Dr. Zoellner,

We are pleased to inform you that your manuscript has been judged scientifically suitable for publication and will be formally accepted for publication once it complies with all outstanding technical requirements.

With kind regards,

Jeffrey Chalmers, Ph.D.

Academic Editor

PLOS ONE

Additional Editor Comments (optional):

Reviewers' comments:

Reviewer's Responses to Questions

**Comments to the Author**

1. Is the manuscript technically sound, and do the data support the conclusions?

Reviewer #1: Yes

2. Has the statistical analysis been performed appropriately and rigorously? 

Reviewer #1: Yes

3. Have the authors made all data underlying the findings in their manuscript fully available?

Reviewer #1: Yes

4. Is the manuscript presented in an intelligible fashion and written in standard English?

Reviewer #1: Yes

5. Review Comments to the Author

Reviewer #1: This study demonstrated the cell-projection pumping (CPP) mechanism in SAOS-2 osteosarcoma cells and human dermal fibroblasts (HDF). FACS analysis validated the increased size and complexity of SAOS-2 with HDF organelles uptake. Meanwhile, SAOS-2 migrated more quickly with high levels of HDF compared to the one with low HDF labeling. This novel discovery suggested that inhibition of exchange would suppress cancer invasion and metastasis, leading to a new cancer therapy.

6. PLOS authors have the option to publish the peer review history of their article (what does this mean?). If published, this will include your full peer review and any attached files.

Reviewer #1: Yes: Yufei Wang

---

## [Editor Report · Acceptance letter]

1 Nov 2019

PONE-D-19-21420 

Increased cell size, structural complexity and migration of cancer cells acquiring fibroblast organelles by cell-projection pumping 

Dear Dr. Zoellner:

I am pleased to inform you that your manuscript has been deemed suitable for publication in PLOS ONE. Congratulations! Your manuscript is now with our production department. 

With kind regards,

on behalf of

Dr. Jeffrey Chalmers 

Academic Editor

PLOS ONE